# Towards Accelerated Model Training via Bayesian Data Selection

**Zhijie Deng**[1]*, **Peng Cui**[2]*, **Jun Zhu**[2]†
[1]Qing Yuan Research Institute, Shanghai Jiao Tong University
[2]Dept. of Comp. Sci. & Tech., Institute for AI, BNRist Center,
Tsinghua-Bosch Joint ML Center, THBI Lab, Tsinghua University, Beijing, 100084 China
`zhijied@sjtu.edu.cn, xpeng.cui@gmail.com, dcszj@tsinghua.edu.cn`

## Abstract

Mislabeled, duplicated, or biased data in real-world scenarios can lead to prolonged training and even hinder model convergence. Traditional solutions prioritizing easy or hard samples lack the flexibility to handle such a variety simultaneously. Recent work has proposed a more reasonable data selection principle by examining the data's impact on the model's generalization loss. However, its practical adoption relies on less principled approximations and additional holdout data. This work solves these problems by leveraging a lightweight Bayesian treatment and incorporating off-the-shelf zero-shot predictors built on large-scale pre-trained models. The resulting algorithm is efficient and easy to implement. We perform extensive empirical studies on challenging benchmarks with considerable data noise and imbalance in the online batch selection scenario, and observe superior training efficiency over competitive baselines. Notably, on the challenging Web-Vision benchmark, our method can achieve similar predictive performance with significantly fewer training iterations than leading data selection methods.

## 1 Introduction

The past year has witnessed significant breakthroughs in deep learning research and applications, with Stable Diffusion [38], ChatGPT [32], and SAM [22] as representative examples. Practitioners have realized that the quality of data used to fuel AI systems is critical in unlocking their full potential. Unfortunately, real-world scenarios often present mislabeled, duplicated, or biased data. As a result, it is paramount to develop methods that can prioritize *valuable* training data to enable more efficient model training and even improved model convergence.

Data selection for accelerating the training of deep models is gaining increasing interest. Some studies, such as curriculum learning, advocate prioritizing *easy* samples in the early training stages [1], but these samples quickly become redundant once they have been learned, making continued training on them a waste of time. On the other hand, online batch selection methods [26, 19, 17] prioritize *hard* samples with high training loss or gradient norm to avoid duplicate training. Nevertheless, in practice, the hardness of samples often arises from pathologies such as improper annotations, inherent ambiguity, or unusual patterns, rendering it problematic to prioritize such samples [31].

The reducible hold-out loss selection (RHO-LOSS) approach [31] addresses these issues by quantifying the usefulness of a sample based on its marginal influence on the model's *generalization* loss, forming a theoretically grounded and universal objective for data selection. However, the estimation of this objective is non-trivial, and RHO-LOSS has to rely on less principled approximations for

---

*Equal contribution.   †Corresponding author.

practical adoption. Besides, RHO-LOSS hinges on a considerable number of *holdout data* to train an auxiliary validation model, which can be costly and should be performed repeatedly for new tasks.

This paper aims to bridge this gap to make the generalization loss-based data selection principle more accessible to a broader audience. We establish a more reasonable approximation of the original objective than RHO-LOSS while eliminating the need for holdout data. To achieve this, we derive a lower bound of the objective to separate the posterior predictive defined on the training data from that defined on the holdout data. Afterward, we propose to use *off-the-shelf* zero-shot predictors, built upon large-scale pre-trained models [35, 41], as a proxy for the latter, since these models often contain generally applicable information for solving specific downstream tasks.

We maintain a Bayesian treatment of the training model to ensure an accurate estimation of the original objective. Bearing in mind that our original goal is to accelerate the training of a *deterministic* model, we adopt the simple and effective Laplace approximation [29, 37, 8] for Bayesian inference. It effortlessly converts point-estimate parameters to a Gaussian posterior. We further introduce Kronecker-factored (KFAC) [30] and last-layer [23] approximations to accelerate the processing of modern neural networks (NNs), resulting in an efficient and easy-to-implement algorithm.

We conduct comprehensive empirical studies on various benchmarks to evaluate the effectiveness of our method. The experiments on standard image recognition tasks demonstrate that our approach can outperform various baselines in aspects of training speed and final accuracy. This conclusion also holds for learning with label noise and class imbalance. On the challenging *WebVision* [25] dataset, which contains plenty of noisy labels and ambiguous images collected from the internet, our method significantly reduces the number of training steps needed to reach the target accuracy and achieves up to *19%* higher final accuracy than prior arts (see Table 3). These results highlight the practical value of our approach. In addition, we conduct informative ablation studies to gain a better understanding of the behavior of our method.

## 2 Background

In this section, we briefly revisit the concept of online batch selection [26] and the data selection principle defined with the model's generalization loss in [31].

Consider training a $\theta$-parameterized deep model $f_\theta : \mathcal{X} \to \mathbb{R}^k$ on a dataset $\mathcal{D} = \{(x_i, y_i)\}_{i=1}^n$ using stochastic gradient descent (SGD). The model likelihood is formally $p(y|x, \theta) = p(y|f_\theta(x))$. At each training step $t$, we can access a data batch $B_t$ of size $n_B$ from $\mathcal{D}$. In online batch selection, we need to compute statistics of the samples in $B_t$ and select only those that meet certain requirements to update the model. This filtering process aims to remove samples that are deemed less valuable.

Let $\mathcal{D}_{t-1}$ denote the set of data observed before $t$ and $(x', y')$ a sample from $B_t$. If we accept selecting $(x', y')$, the updated predictive distribution, in a Bayesian view, will be $p(y|x, \mathcal{D}_{t-1}, \{(x', y')\}) = \mathbb{E}_{p(\theta|\mathcal{D}_{t-1}, \{(x', y')\})} p(y|x, \theta)$.[2] The question arises how to estimate the quality of this distribution to determine which sample $(x', y') \in B_t$ should be selected. A natural tactic is to compare this distribution to the ground truth data-generating distribution $\dot{p}(x, y)$. That said, the typical KL divergence can be introduced, and our goal becomes solving the following problem:

$$\min_{(x', y') \in B_t} \mathbb{E}_{\dot{p}(x)} \big( D_{\text{KL}} \big[ \dot{p}(y|x) \| p(y|x, \mathcal{D}_{t-1}, \{(x', y')\}) \big] \big) = \text{const.} - \mathbb{E}_{\dot{p}(x,y)} \big[ \log p(y|x, \mathcal{D}_{t-1}, \{(x', y')\}) \big],$$
(1)

where $\text{const.}$ denotes a constant agnostic to the object to optimize. By applying Monte Carlo (MC) estimation using extra holdout samples $\mathcal{D}^* = \{(\tilde{x}_i, \tilde{y}_i)\}_{i=1}^m$ from $\dot{p}(x, y)$, we arrive at the following optimization problem:

$$\max_{(x', y') \in B_t} \frac{1}{m} \sum_{i=1}^m \big[ \log p(\tilde{y}_i | \tilde{x}_i, \mathcal{D}_{t-1}, \{(x', y')\}) \big] \iff \max_{(x', y') \in B_t} \log p(\mathcal{D}^* | \mathcal{D}_{t-1}, \{(x', y')\}). \quad (2)$$

This objective corresponds to the model's *generalization* loss instead of the fitness of training data.

By Bayes' rule, there is:

$$p(\mathcal{D}^* | \mathcal{D}_{t-1}, \{(x', y')\}) = \frac{p(y'|x', \mathcal{D}^*, \mathcal{D}_{t-1}) p(x', \mathcal{D}^*, \mathcal{D}_{t-1})}{p(y'|x', \mathcal{D}_{t-1}) p(x', \mathcal{D}_{t-1})} = \frac{p(y'|x', \mathcal{D}^*, \mathcal{D}_{t-1})}{p(y'|x', \mathcal{D}_{t-1})} \cdot p(\mathcal{D}^* | \mathcal{D}_{t-1}, x'), \quad (3)$$

---

[2]We assume selecting one single sample per time for simplicity. Multi-sample selection is viable.

where the item $p(\mathcal{D}^*|\mathcal{D}_{t-1}, x')$ actually equals to the constant $p(\mathcal{D}^*|\mathcal{D}_{t-1})$ because $x'$ alone cannot incur model update. Plugging this back into Equation (2), we arrive at the final objective for data selection:

$$\max_{(x,y)\in B_t} \log p(y|x, \mathcal{D}^*, \mathcal{D}_{t-1}) - \log p(y|x, \mathcal{D}_{t-1}), \qquad (4)$$

where we omit constants and abuse $(x, y)$ to represent $(x', y')$ hereinafter if there is no misleading.

Although the above selection principle is theoretically sound and universally applicable, estimating it accurately is challenging. In particular, it is difficult to estimate the posterior predictive distribution defined with the combination of training and holdout data in a computationally efficient manner. To address this issue, RHO-LOSS proposes approximating $\log p(y|x, \mathcal{D}^*, \mathcal{D}_{t-1})$ with $\log p(y|x, \mathcal{D}^*)$ and approximating the posterior predictive with the point-estimate model' predictions [31]. However, these approximations compromise the method's theoretical groundness, and access to holdout data can often be infeasible in practice. Our approach overcomes these limitations by utilizing a lightweight Bayesian treatment and incorporating off-the-shelf zero-shot predictors built on large-scale pre-trained models, as detailed below.

## 3 Methodology

In this section, we demonstrate that the data selection principle discussed earlier can be lower-bounded by more easily computable quantities. We then leverage these insights to develop a Bayesian data selection approach to accelerating the training of deterministic deep models.

### 3.1 The Lower Bound

As previously discussed, we need to estimate two log-probabilities, $\log p(y|x, \mathcal{D}^*, \mathcal{D}_{t-1})$ and $\log p(y|x, \mathcal{D}_{t-1})$, for each sample in $B_t$ to identify the most useful one and update the model accordingly. However, as mentioned, evaluating the former is challenging. To address this issue, we derive a lower bound that allows for a more feasible estimation.

We first unfold $\log p(y|x, \mathcal{D}^*, \mathcal{D}_{t-1})$ as the combination of a posterior $p(\theta|\mathcal{D}^*, \mathcal{D}_{t-1})$ and the model likelihood $p(y|x, \theta)$, detailed below (we defer the derivation to Appendix):

$$\log p(y|x, \mathcal{D}^*, \mathcal{D}_{t-1}) = \log \int p(\mathcal{D}^*|\theta)p(\theta|\mathcal{D}_{t-1})p(y|x, \theta)d\theta - \log p(\mathcal{D}^*|\mathcal{D}_{t-1}). \qquad (5)$$

Then, by Jensen's inequality, there is (derivation is deferred to Appendix)

$$\log p(y|x, \mathcal{D}^*, \mathcal{D}_{t-1}) \geq \mathbb{E}_{p(\theta|\mathcal{D}_{t-1})} \log p(y|x, \theta) + \mathbb{E}_{p(\theta|\mathcal{D}_{t-1})} \log p(\mathcal{D}^*|\theta) - \log p(\mathcal{D}^*|\mathcal{D}_{t-1}). \qquad (6)$$

Notably, the last two terms are independent of $(x, y)$ and can be excluded from the optimization.

We can similarly derive another lower bound:

$$\log p(y|x, \mathcal{D}^*, \mathcal{D}_{t-1}) = \log \int p(\mathcal{D}_{t-1}|\theta)p(\theta|\mathcal{D}^*)p(y|x, \theta)d\theta - \log p(\mathcal{D}_{t-1}|\mathcal{D}^*)$$
$$\geq \mathbb{E}_{p(\theta|\mathcal{D}^*)} \log p(y|x, \theta) + \mathbb{E}_{p(\theta|\mathcal{D}^*)} \log p(\mathcal{D}_{t-1}|\theta) - \log p(\mathcal{D}_{t-1}|\mathcal{D}^*), \qquad (7)$$

where the last two terms are also agnostic to $(x, y)$ and hence can be omitted.

Combining Equation (6) and Equation (7), there is

$$\log p(y|x, \mathcal{D}^*, \mathcal{D}_{t-1}) \geq \alpha \mathbb{E}_{p(\theta|\mathcal{D}_{t-1})} \log p(y|x, \theta) + (1 - \alpha)\mathbb{E}_{p(\theta|\mathcal{D}^*)} \log p(y|x, \theta) + \text{const.}, \quad (8)$$

where $\alpha \in [0, 1]$ represents a trade-off coefficient. This way, we disentangle the posterior associated with the training data $\mathcal{D}_{t-1}$ from that associated with the holdout data $\mathcal{D}^*$.

Rewriting $p(y|x, \mathcal{D}_{t-1})$ as $\mathbb{E}_{p(\theta|\mathcal{D}_{t-1})}p(y|x, \theta)$, we can subsequently convert Equation (4) to the following maximization problem:

$$\max_{(x,y)\in B_t} \alpha \mathbb{E}_{p(\theta|\mathcal{D}_{t-1})} \log p(y|x, \theta) + (1 - \alpha)\mathbb{E}_{p(\theta|\mathcal{D}^*)} \log p(y|x, \theta) - \log \mathbb{E}_{p(\theta|\mathcal{D}_{t-1})}p(y|x, \theta). \quad (9)$$

The presented objective is intriguing for several reasons. Firstly, the first term and $\alpha$ times the third term perform expectation and logarithm operations in reverse order, resulting in a quantity similar to

the uncertainty estimates in Bayesian deep learning [39] (nonetheless, our objective involves data labels). Additionally, the third term helps to prevent the selection of redundant samples. And, the second term prioritizes points with high semantic alignment with their annotations. These three forces are integrated adaptively to accelerate model training across various stages.

## 3.2 Zero-shot Predictor as the Validation Model

Collecting extra holdout data to train an auxiliary validation model can be costly and should be performed repeatedly for new tasks. To address this issue, we propose to use zero-shot predictors built upon large-scale pre-trained models [35, 41] as a proxy for the validation model, based on the observation that they usually exhibit promising transfer performance across a broad range of downstream applications.

Formally, we make the following approximation:

$$\mathbb{E}_{p(\theta|\mathcal{D}^*)} \log p(y|x, \theta) \approx \log p(y|\tilde{f}(x)), \tag{10}$$

where $\tilde{f} : \mathcal{X} \to \mathbb{R}^k$ denotes the zero-shot predictor used.

We can think of the pre-trained model as a universal validation model trained on an extensive dataset, leading to the Bayesian posterior collapsing to a point estimate. Although its training data may not precisely follow the data-generating distribution for the current task, they share fundamental patterns with the data in our problem, making the above approximation reasonable. Notably, as shown in Section 4, our trained model eventually performs much better than the zero-shot predictor.

## 3.3 Lightweight Bayesian Treatment of the Training Model

The first and third terms in Equation (9) raise the requirement of maintaining a Bayesian posterior $p(\theta|\mathcal{D}_{t-1})$ during training. Due to the high nonlinearity of deep NNs, the analytical posterior is intractable. By convention, we can introduce an approximate posterior $q(\theta|\mathcal{D}_{t-1})$ and tune it by approximate Bayesian inference methods like MCMC [42, 5, 45], variational inference [2, 15, 28, 44, 21], Laplace approximation [29, 37], etc. Recalling that, despite relying on a Bayesian treatment, our final goal is to accelerate and improve the training of a *deterministic* model, Laplace approximation is well suited to our setting—it can convert point-estimate parameters to a Gaussian posterior effortlessly, thus lifting the unnecessary burden of learning and maintaining a posterior explicitly. We clarify that, technically, other Bayesian inference methods are also compatible with our approach.

Although Laplace approximation is typically used for maximum a posteriori estimation, it can be effectively adapted to the online learning cases [36]. Specifically, consider an isotropic Gaussian prior with precision $\tau_0$ over parameters $\theta$. Let $\mathcal{D}_{t-1} := b_1 \cup b_2 \cup \cdots \cup b_{t-1}$[3] denote all selected training samples before time $t$. The parameters of our deterministic model at time $t - 1$, dubbed as $\theta_{t-1}$, are likely to approach a mode of the true posterior $p(\theta|\mathcal{D}_{t-1})$ in the presence of a proper weight decay regularizer in stochastic optimization [10]. The online Laplace approximation then deploys the following approximate posterior:

$$q(\theta|\mathcal{D}_{t-1}) = \mathcal{N}(\theta_{t-1}, H_{t-1}^{-1}), \ H_{t-1} = \tau_0 I + \sum_{i=1}^{t-1} \Big( \sum_{x,y \in b_i} \frac{\partial^2 [-\log p(y|x, \theta)]}{\partial \theta^2}\Big|_{\theta=\theta_i} \Big). \tag{11}$$

The Gaussian covariance should be positive semi-definite, but as $\theta_{t-1}$ cannot be guaranteed to be exactly the posterior mode, we would better replace the Hessian matrix with the generalized Gauss-Newton (GGN) matrix to avoid ill-posed approximation, resulting in:

$$q(\theta|\mathcal{D}_{t-1}) = \mathcal{N}(\theta_{t-1}, G_{t-1}^{-1}), \ G_{t-1} = \tau_0 I + \sum_{i=1}^{t-1} \Big( \sum_{x,y \in b_i} J_{\theta_i}(x)^\top \Lambda_{\theta_i}(x, y) J_{\theta_i}(x) \Big), \tag{12}$$

where $J_{\theta_i}(x) := \nabla_\theta f_\theta(x)|_{\theta=\theta_i}$ and $\Lambda_{\theta_i}(x, y) := \nabla_f^2 [-\log p(y|f)]|_{f=f_{\theta_i}(x)}$.

**Practical acceleration.** Modern neural networks usually contain millions of parameters, making the matrix $G_{t-1}$ too large to fit into CPU/GPU memories. To address this, a common practice is to sparsify the matrix using diagonal or Kronecker-factored (KFAC) approximations [30]. This work

---

[3] $b_i \subset B_i$ represents the set of selected samples at time step $i$.

**Algorithm 1** Bayesian data selection to accelerate the training of deterministic deep models.

---

1: **Input:** Number of iterations $T$, dataset $\mathcal{D}$, prior precision $\tau_0$, number of effective data $n_e$, batch size $n_B$, number of selections $n_b$, zero-shot predictor $\tilde{f}$, deterministic model with parameters $\theta$.
2: Intialize $\theta_0$, $A_0 \leftarrow 0$, $G_0 \leftarrow 0$;
3: **for** $t$ in $1, \ldots, T$ **do**
4:     Draw a mini-batch $B_t$ from $\mathcal{D}$;
5:     $V_{t-1} \leftarrow \sqrt{n_e}A_{t-1} + \sqrt{\tau_0}I$, $U_{t-1} \leftarrow \sqrt{n_e}G_{t-1} + \sqrt{\tau_0}I$;
6:     Estimate the objective in Equation (16) for every sample in $B_t$ and select the top-$n_b$ ones to form $b_t$;
7:     Perform back-propagation with $\sum_{x,y \in b_t} \log p(y|f_{\theta_{t-1}}(x))$;
8:     Apply weight decay regularization and do gradient ascent to obtain $\theta_t$;
9:     Use the last-layer features and softmax gradients to update $A_t$ and $G_t$ with exponential moving average;
10: **end for**

---

prefers KFAC as it preserves the correlations between parameters within the same layers. Besides, a recent surprising finding is that we can even apply Laplace approximation to only the last layer of a deep NN, leaving the other parameters point-estimate, to conjoin efficiency and calibrated uncertainty estimates [23, 8]. As a result, we consider combining last-layer and KFAC approximations to reduce the burden caused by the Bayesian treatment, with the details presented below.

Let us decompose $f_{\theta_i}$ as a feature extractor $h_{\theta_i}$ and a linear layer with weights $\theta_i^{(l)} \in \mathbb{R}^{d \times k}$, i.e., $f_{\theta_i}(x) := h_{\theta_i}(x)^\top \theta_i^{(l)}$. We adapt the GGN matrix associated with the last-layer weights $\theta_i^{(l)}$ derived in [37, 23] to the following formula:

$$G_{t-1}^{(l)} \approx V_{t-1} \otimes U_{t-1} := (\sqrt{|\mathcal{D}_{t-1}|}A_{t-1} + \sqrt{\tau_0}I) \otimes (\sqrt{|\mathcal{D}_{t-1}|}G_{t-1} + \sqrt{\tau_0}I), \qquad (13)$$

where $\otimes$ denotes the Kronecker product and

$$A_{t-1} := \frac{1}{|\mathcal{D}_{t-1}|} \sum_{i=1}^{t-1} \sum_{x,y \in b_i} h_{\theta_i}(x)h_{\theta_i}(x)^\top,$$

$$G_{t-1} := \frac{1}{|\mathcal{D}_{t-1}|} \sum_{i=1}^{t-1} \sum_{x,y \in b_i} [\nabla_f \log p(y|f)|_{f=f_{\theta_i}(x)}][\nabla_f \log p(y|f)|_{f=f_{\theta_i}(x)}]^\top. \qquad (14)$$

Of note that the matrices $A_{t-1} \in \mathbb{R}^{d \times d}$ and $G_{t-1} \in \mathbb{R}^{k \times k}$ raise only minimal extra storage cost. The approximate posterior can be formulated as a matrix-variate Gaussian [12]: $q(\theta^{(l)}|\mathcal{D}_{t-1}) = \mathcal{MN}(\theta^{(l)}|\theta_{t-1}^{(l)}, U_{t-1}^{-1}, V_{t-1}^{-1})$. Given the linear nature, it is straightforward to get the distribution over the model output $f_x$ for input $x$ (derivation is deferred to Appendix):

$$q(f_x|\mathcal{D}_{t-1}) = \mathcal{N}\left(f_{\theta_{t-1}}(x), \left(h_{\theta_{t-1}}(x)^\top V_{t-1}^{-1} h_{\theta_{t-1}}(x)\right) U_{t-1}^{-1}\right). \qquad (15)$$

**The final data selection objective.** With the above equation, the selection objective boils down to

$$\max_{(x,y) \in B_t} \alpha\left[\frac{1}{S} \sum_{s=1}^{S} \log p(y|f_x^{(s)})\right] + (1-\alpha)\log p(y|\tilde{f}(x)) - \log\left[\frac{1}{S} \sum_{s=1}^{S} p(y|f_x^{(s)})\right], \qquad (16)$$

where $f_x^{(s)} \sim q(f_x|\mathcal{D}_{t-1})$, $s = 1, \ldots, S$ are MC samples. Compared to the non-last-layer Laplace approximation, which involves sampling over the parameter space and performing MC integration, the last-layer one enjoys a much faster evaluation procedure. It also enables the use of a large $S$. Our method can also abandon the KFAC approximation when the linear head is small.

**The algorithm.** We present the training procedure in Algorithm 1. To avoid too large $|\mathcal{D}_{t-1}|$ and hence too sharpened approximate posterior, we use a tunable hyper-parameter $n_e$ to replace $|\mathcal{D}_{t-1}|$ in the computation of $V_{t-1}$ and $U_{t-1}$. For implemental simplicity, we use the exponential moving average technique to update $A_t$ and $G_t$. For models equipped with batch normalization layers [16], we perform an extra forward propagation for the selected data $b_t$ to obtain proper batch statistics for model update, following [31]. Our method takes a similar amount of time per iteration as [31]. The primary difference is that we use a zero-shot predictor based on pre-trained models to compute the second term in Equation (16), whereas [31] uses a validation model. Computing the Gaussian covariance in Equation (15), MC sampling, and updating $A_t$ and $G_t$ consume neglectable resources.

Table 1: Epochs needed to reach a target test accuracy on clean and noisy data (final accuracy is reported in parentheses). CIFAR-10*/100* denotes adding 10% symmetric label noise to the dataset. Best performance is highlighted in **bold**. "-" indicates that the target accuracy was not reached. For all methods, only half of the original training set is used for training. The target accuracies are set following RHO-LOSS [31].

| Method\Dataset | CIFAR-10 | | CIFAR-10* | | CIFAR-100 | | CIFAR-100* | |
|---|---|---|---|---|---|---|---|---|
| CLIP Acc | 75.6% | | 75.6% | | 41.6% | | 41.6% | |
| Target Acc | 80.0% | 87.5% | 75.0% | 85.0% | 40.0% | 52.5% | 40.0% | 47.5% |
| Train Loss | 81 | 129 (90%) | - | - (28%) | 138 | - (42%) | - | - (4%) |
| Grad Norm | - | - (61%) | - | - (23%) | 139 | - (42%) | - | - (4%) |
| Grad Norm IS | 57 | 139 (89%) | 57 | - (84%) | 71 | 132 (55%) | 94 | 142 (48%) |
| SVP | - | - (55%) | - | - (48%) | - | - (18%) | - | - (14%) |
| Irred Loss | - | - (60%) | - | - (62%) | 93 | - (43%) | 89 | - (43%) |
| Uniform | 79 | - (87%) | 62 | - (85%) | 65 | 133 (54%) | 79 | 116 (50%) |
| RHO-LOSS | 39 | 65 (91%) | 27 | 49 (91%) | 48 | 77 (61%) | 49 | 65 (60%) |
| Proposed | **33** | **61 (91%)** | **25** | **47 (91%)** | **32** | **53 (63%)** | **39** | **53 (61%)** |

# 4 Experiment

We compare the proposed method to the prior state-of-the-art (SOTA) data selection methods on various image classification benchmarks, including standard datasets (e.g., CIFAR-10 and CIFAR-100 [24]), their variants with controllable degrees of label noise and class-imbalance, and a real-world, large-scale, noisy, and imbalanced dataset, WebVision [25]. We also diagnose our selection strategy through elaborate ablation studies.

**Datasets.** We first evaluate the proposed method on clean CIFAR-10/100 and noisy CIFAR-10/100 with 10% symmetric label noise. Furthermore, we experiment in the context of imbalanced datasets, specifically CIFAR-10/100 datasets with varying degrees of class-imbalance ratio. Last, we investigate a web-scraped and large-scale dataset – WebVision, which consists of over $2.5$ million images in $1000$ categories and suffers from severe label noise and class-imbalance issues. For a fair comparison with RHO-LOSS [31], only half of the training set is used for model training.

**Baselines.** We compare the proposed method to a variety of baselines that define selection principles with various metrics, including the (training) loss [20], gradient norm [19], and gradient norm with importance sampling (gradient norm IS) [19]. We also compare to uniform sampling, Selection-via-Proxy (SVP) [6] and the latest SOTA method, RHO-LOSS [31].

**Implementation details.** In experiments under label noise, we introduce 10% symmetric noise caused by randomly flipping the ground-truth label to other labels. For the class-imbalance setting, we consider the long-tailed imbalance with two levels of imbalance intensity (i.e., 10 and 100), where an exponential decay on sample size is introduced across classes [3]. We use the same optimizer (AdamW [27]) and hyperparameters (e.g., learning rate 0.001, weight decay of 0.01, $n_b = 32$ and $\frac{n_b}{n_B} = 0.1$) as RHO-LOSS. Unless specified otherwise, we use ResNet18 (RN18) [14] as the deterministic model and specify the zero-shot predictor with CLIP-RN50. We select the trade-off coefficient $\alpha$ from $\{0.1, 0.2, 0.3, 0.4\}$ and the number of effective data $n_e$ from $\{100, 200, 500, 1000\}$. In most cases, we set the prior precision $\tau_0$ to 1. We report average results over 3 random runs.

**Evaluation.** Following [31], we evaluate speedup by comparing the number of epochs required to reach a given target test accuracy, which implicitly reflects the quality of the selection strategy.

## 4.1 Speedup on Clean Data and Noisy Data

We first empirically evaluate the proposed method on CIFAR-10 and CIFAR-100 with/without label noise. Table 1 reports the results. We can observe that the proposed method achieves considerably improved training speed compared to competing baselines in both scenarios with and without label noise. This evidences the efficacy of our Bayesian selection method in filtering out redundant and noisy data points in the training set. Despite leveraging extra holdout data, RHO-LOSS still

Table 2: Epochs required to reach a target test accuracy on imbalanced CIFAR-10 and CIFAR-100 (final accuracy is reported in parentheses).

| Dataset | Imbalance Ratio | Target Acc | Uniform | RHO-LOSS | Proposed |
|---------|-----------------|------------|---------|----------|----------|
| CIFAR-10 | 10 | 60% | 68 | 36 | **34** |
| | | 70% | 98 (75%) | 56 (80%) | **52 (83%)** |
| | 100 | 50% | 81 | 50 | **44** |
| | | 60% | 119 (56%) | 87 (62%) | **79 (68%)** |
| CIFAR-100 | 10 | 20% | 69 | 34 | **32** |
| | | 30% | 110 (33%) | 62 (39%) | **58 (45%)** |
| | 100 | 15% | 70 | 43 | **41** |
| | | 20% | - (19%) | 71 (24%) | **66 (28%)** |

Table 3: Epochs required to reach a target test accuracy on WebVision dataset (final accuracy is reported in parentheses).

| Dataset | Validation set | Target Acc | Uniform | RHO-LOSS | Proposed |
|---------|----------------|------------|---------|----------|----------|
| WebVision-100 | WebVision | 30% | - | 29 | **25** |
| | | 40% | - (27%) | 47 (42%) | **40 (60%)** |
| | ILSVRC12 | 30% | - | 40 | **33** |
| | | 40% | - (23%) | - (37%) | **50 (55%)** |
| WebVision-200 | WebVision | 30% | - | 28 | **22** |
| | | 40% | - (26%) | 48 (42%) | **36 (61%)** |
| | ILSVRC12 | 30% | - | 35 | **29** |
| | | 40% | - (19%) | - (39%) | **46 (56%)** |

underperforms our approach, potentially due to the less reliable approximations. Surprisingly, the superiority of our method is even more significant on the more challenging CIFAR-100. We also point out that our method eventually reaches higher final accuracy than the zero-shot CLIP. In other words, our method does not hinge on a performant zero-shot predictor.

## 4.2 Speedup on Class-imbalance Data

We further evaluate the proposed method on class-imbalanced data, a typical pattern in the real world while probably raising non-trivial difficulties for model training [7, 3]. Specifically, we experiment on the aforementioned imbalanced CIFAR-10 and CIFAR-100 using similar protocols to previous studies. The test datasets remain unchanged. Given that RHO-LOSS is superior to other baselines, we primarily compare with it in the following studies. We also add uniform sampling into comparison due to its implementation simplicity. Table 2 presents the results.

As shown, the proposed method still consistently outperforms uniform sampling and RHO-LOSS. In particular, the final accuracy achieved by our method is higher than RHO-LOSS. We also note that the gain in training efficiency becomes more significant as the imbalance ratio increases. The failure of RHO-LOSS is partially attributed to the less principled approximations. Another reason is that the holdout data are still class-imbalanced, rendering the trained validation models biased.

## 4.3 Speedup on Large Web-scraped Data

Learning algorithms face significant challenges when dealing with noisy and imbalanced real-world datasets. To evaluate our proposed method's efficacy in addressing these challenges, we assess it on a large-scale web-scraped dataset, WebVision. We defer the details of WebVision to the Appendix. Since the dataset is large, comprising 2.4 million data points, we train classification models only on the first 100/200 classes (known as WebVision-100/200) of the entire dataset, following [18, 4], for an efficient evaluation. We test the trained models on the human-annotated WebVision validation set

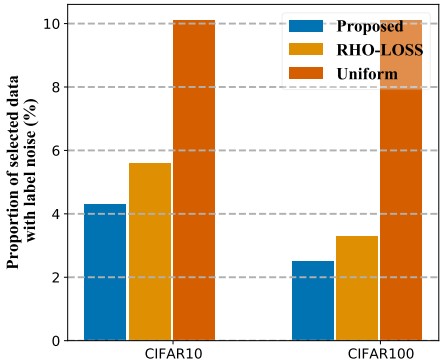

(a) Proportion of label noise in selection.

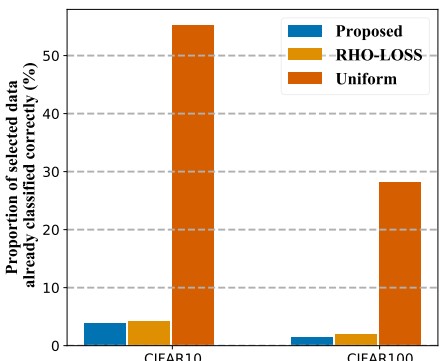

(b) Proportion of redundant points in selection.

Figure 1: Properties of the data selected by our method and baselines on CIFAR-10 and CIFAR-100 with $10\%$ label noise. Redundant points represent the data that have already been classified correctly. The reported values are averaged over 150 epochs of model training and five random runs.

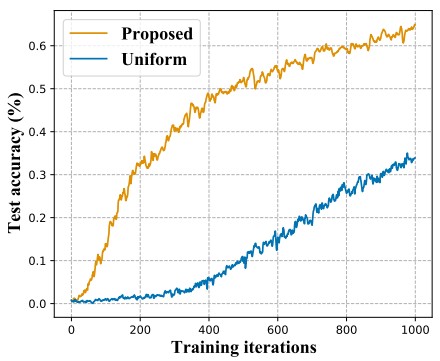

Figure 2: Training curves corresponding to using pre-trained ViT-B/16 as the model backbone. (WebVision-200; 1 epoch=344 iterations)

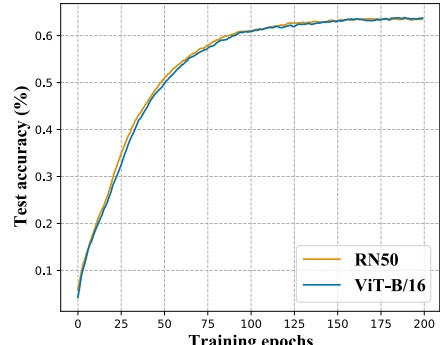

Figure 3: Training curves corresponding to using zero-shot predictors defined with various CLIP backbones. (CIFAR-100)

and the ILSVRC12 validation set [9]. We report the results in Table 3. We select 30% and 40% as target accuracies according to the final accuracies of RHO-LOSS ($\sim$40%) and our approach ($\sim$60%).

Still, the proposed method outperforms uniform sampling and RHO-LOSS in aspects of both the speed to reach the target accuracy and the final accuracy on both WebVision-100 and WebVision-200. Notably, our method achieves up to *19%* higher final accuracy than RHO-LOSS, suggesting the superiority of our method in coping with real-world datasets. We also report results for our method on the entire training set in the Appendix, which shows more significant speedups and higher final accuracy due to more training data.

## 4.4 Analysis of the Properties of the Selected Data

In this section, we analyze the characteristics of the data selected by our method and compare it with existing works. Figure 1a displays the proportion of selected data points with label noise, and it is evident that data selected by our method has the lowest label noise. We also investigate whether our method prioritizes redundant data, defined as those having already been correctly classified by the training model [31]. As depicted in Figure 1b, both our method and RHO-LOSS select significantly fewer redundant data points than uniform sampling, and our method is slightly better.

## 4.5 Ablation Studies

In this section, we conduct ablation studies on model architecture, zero-shot predictor, and some critical hyper-parameters (e.g., $\alpha$, $n_e$).

Table 5: The efficacy of a variant of our method (referred to as Proposed†) where the zero-shot validation model is replaced by that used by RHO-LOSS [31]. Experiment on CIFAR-100.

|  | RHO-LOSS | Proposed† | Proposed |
|---|---|---|---|
| Epochs to reach 40.0% test accuracy | 48 | 30 | 32 |
| Epochs to reach 52.5% test accuracy | 77 | 52 | 53 |
| Final accuracy | 61% | 63% | 63% |

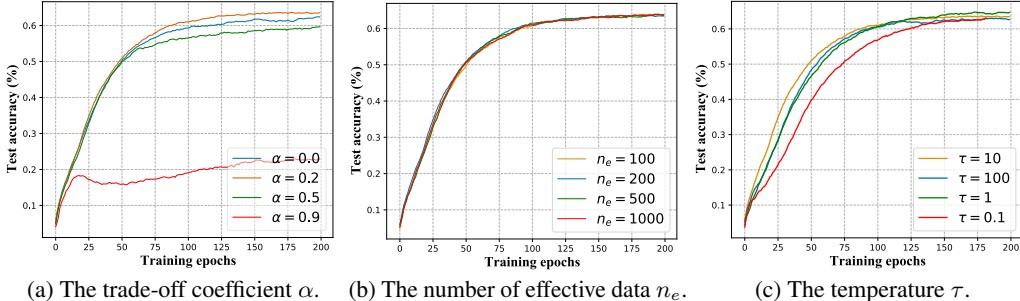

(a) The trade-off coefficient $\alpha$.  (b) The number of effective data $n_e$.  (c) The temperature $\tau$.

Figure 4: Ablation studies on some critical hyper-parameters including $\alpha$, $n_e$, and $\tau$ on CIFAR-100.

**Model architecture.** We test our method using the powerful Vision Transformer (ViT) [11] architecture on WebVision-200. Specifically, we experiment with a ViT-B/16 model pre-trained unsupervisedly by Masked Auto-encoder [13], considering that fine-tuning a pre-trained model is preferred over training a model from scratch in practice. We provide the training curve of our method in Figure 2, where uniform sampling is included for comparison. The huge performance gap between our method and the baseline validates the efficacy of our method in this setting.

**Zero-shot predictor.** We next assess the impact of the zero-shot predictor on our method. We specify the zero-shot predictor with various CLIP variants (e.g., RN50 and ViT-B/16) and draw the training curves in Figure 3. As shown, although the zero-shot accuracy of the RN50 variant is significantly lower than that of ViT-B/16, the speedup effect of the two backbones is similar. This finding suggests the robustness of our method against the selection of the zero-shot predictor.

In the above experiments, we use CLIP-based zero-shot predictors without further tuning. Here, we perform linear probing using the CLIP models to demonstrate if we can achieve good convergence trivially. We simply adopt the uniform sampling strategy and report the results in Table 4. We do not report convergence speed because the baseline uses pre-trained weights while our method trains models from scratch. As shown, our method still bypasses the baseline with clear margins.

Table 4: Comparison of final accuracy (%) between the proposed method and the linear probing with CLIP and uniform sampling strategy.

| Dataset | CLIP linear probing | Proposed |
|---|---|---|
| CIFAR-10 | 84.5 | 91.4 |
| CIFAR-10* | 84.1 | 91.3 |
| CIFAR-100 | 58.5 | 63.3 |
| CIFAR-100* | 57.8 | 61.4 |

We replace the zero-shot predictor used in our method with the validation model in RHO-LOSS [31] and present the corresponding results in Table 5. The comparison proves the superiority of our method over RHO-LOSS. These results also reflect that our method is robust against the specification of the validation model.

**Hyper-parameters.** We then analyze the effects of three crucial hyper-parameters, $\alpha$, $n_e$, and the temperature $\tau$ used by the softmax operation in the CLIP zero-shot predictor. Figure 4a shows training curves associated with various $\alpha$. We see that too large $\alpha$ (i.e., lightening the second term in Equation (9)) can lead to a significant drop in training speed and final performance, which emphasizes the importance of the zero-shot predictor. Nevertheless, using only the zero-shot predictor (i.e., $\alpha = 0$) is also suboptimal, leading to degraded final accuracy. Hence, we conclude that both terms in the lower bound of the original data selection principle are beneficial, where the first term accounts more for the later training stages while the second term accounts more for the earlier ones. Figure 4b shows the ablations study on $n_e$, and we witness the robustness of our method against $n_e$. Figure 4c

shows training curves corresponding to different $\tau$. We observe that an appropriate temperature boosts our method considerably.

**Wall-clock time.** We empirically observe that the per-epoch training time for Uniform, RHO-LOSS, and our method on CIFAR-100 is 14s, 18s, and 21s, respectively. Namely, our approach has slightly increased per-epoch time over RHO-LOSS. It arises from that we use a CLIP-RN50 zero-shot predictor to compute the second term in Equation (16), whereas [31] uses a RN18 validation model. In fact, this gap can be reduced by a simple implementation trick—pre-computing the CLIP-RN50 predictions for each sample in the training set before training.

As shown in Table 1, compared to RHO-LOSS, we can reduce the number of epochs required to reach 40% test accuracy from 48 to 32 and 52.5% from 77 to 53. According to the per-epoch training time, we achieve a $48 * 18/32/21 \approx 1.29$ or $77 * 18/53/21 \approx 1.25$ times practical acceleration.

## 5    Related Works

Extensive methods have been proposed to accelerate model training through techniques such as data pruning, coreset selection, curriculum learning, online batch selection, etc. Data pruning explores various metrics, such as EL2N score [33], forgetting score [40], and classification margin [34], to measure individual differences among data points and retains only the hardest examples for model training. However, data pruning still exhibits limitations when dealing with noisy labels, and some of these metrics are computationally expensive. Coreset selection methods also partially address the problem of accelerating model training. In particular, [43] contributes a data scoring mechanism that is robust to the change of scenarios for coreset selection, and [46] makes an in-depth understanding of the catastrophic accuracy drop issue of one-shot coreset selection and contributes a novel solution to it. However, these methods lack the flexibility to prioritize samples with different properties at various training stages. Curriculum learning, as advocated by [1], prioritizes easy points with low label noise before uniformly training on all data points. While this strategy enhances convergence, it fails to address the issue of skipping redundant points already learned.

Online batch selection methods [26, 19, 17] tackle the training acceleration problem by selecting hard data identified by high loss or gradient norm. However, they also suffer from a common drawback—high loss can be caused by label noise or ambiguous labels, so prioritizing such data can result in a decline in predictive performance. Compared to the prior art, our method establishes a Bayesian data selection metric and exploits zero-shot predictors to prioritize valuable training data for addressing these issues.

## 6    Conclusion

This paper addresses the challenges posed by noisy and biased data in real-world scenarios. Our main contribution is to make the generalization loss-based data selection principle more accessible to accelerate the training of deep models. To achieve this, we first derive a lower bound of this objective to improve its tractability. We then propose leveraging a Bayesian treatment and off-the-shelf zero-shot predictors to estimate the data selection objective reliably. The resulting algorithm does not require extra holdout data. Our extensive empirical studies demonstrate the superiority of our method in accelerating model training over competitive baselines.

**Limitation.** Our method may fail when the zero-shot predictor performs poorly on specific tasks. Future work could explore adapting the zero-shot predictor to a few-shot one using a few clean validation data to address this limitation.

## Acknowledgments

Z.J. Deng was supported by Natural Science Foundation of Shanghai (No. 23ZR1428700) and the Key Research and Development Program of Shandong Province, China (No. 2023CXGC010112). J. Zhu and P. Cui were supported by NSF of China Projects (Nos. 62061136001, 61620106010, 62076145, U19B2034, U1811461, U19A2081, 6197222); a grant from Tsinghua Institute for Guo Qiang; and the High Performance Computing Center, Tsinghua University. J.Z was also supported by the XPlorer Prize.

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

# A Proofs

## A.1 Derivation of Equation (5)

$$\log p(y|x, \mathcal{D}^*, \mathcal{D}_{t-1}) = \log \int p(\theta|\mathcal{D}^*, \mathcal{D}_{t-1})p(y|x,\theta)d\theta$$

$$= \log \int \frac{p(\mathcal{D}^*|\theta)p(\theta|\mathcal{D}_{t-1})}{p(\mathcal{D}^*|\mathcal{D}_{t-1})}p(y|x,\theta)d\theta$$

$$= \log \int p(\mathcal{D}^*|\theta)p(\theta|\mathcal{D}_{t-1})p(y|x,\theta)d\theta - \log p(\mathcal{D}^*|\mathcal{D}_{t-1}).$$

## A.2 Derivation of Equation (6)

$$\log p(y|x, \mathcal{D}^*, \mathcal{D}_{t-1}) = \log \int p(\mathcal{D}^*|\theta)p(\theta|\mathcal{D}_{t-1})p(y|x,\theta)d\theta - \log p(\mathcal{D}^*|\mathcal{D}_{t-1})$$

$$\geq \int p(\theta|\mathcal{D}_{t-1}) \log[p(\mathcal{D}^*|\theta)p(y|x,\theta)]d\theta - \log p(\mathcal{D}^*|\mathcal{D}_{t-1})$$

$$= \mathbb{E}_{p(\theta|\mathcal{D}_{t-1})} \log p(\mathcal{D}^*|\theta) + \mathbb{E}_{p(\theta|\mathcal{D}_{t-1})} \log p(y|x,\theta) - \log p(\mathcal{D}^*|\mathcal{D}_{t-1}).$$

## A.3 Derivation of Equation (15)

Given that the posterior of the weights of the last layer is $q(\theta^{(l)}|\mathcal{D}_{t-1}) = \mathcal{MN}(\theta^{(l)}|\theta_{t-1}^{(l)}, U_{t-1}^{-1}, V_{t-1}^{-1})$ and the input to the last layer is $h_{\theta_{t-1}}(x)$, there is

$$q(f_x|\mathcal{D}_{t-1}) = \mathcal{MN}\left(h_{\theta_{t-1}}(x)^\top \theta_{t-1}^{(l)}, U_{t-1}^{-1}, h_{\theta_{t-1}}(x)^\top V_{t-1}^{-1} h_{\theta_{t-1}}(x)\right)$$

$$= \mathcal{N}\left(f_{\theta_{t-1}}(x), \left(h_{\theta_{t-1}}(x)^\top V_{t-1}^{-1} h_{\theta_{t-1}}(x)\right)U_{t-1}^{-1}\right).$$

The second equation stems from that $(h_{\theta_{t-1}}(x)^\top V_{t-1}^{-1} h_{\theta_{t-1}}(x))$ is a scalar.

# B Experiment Details

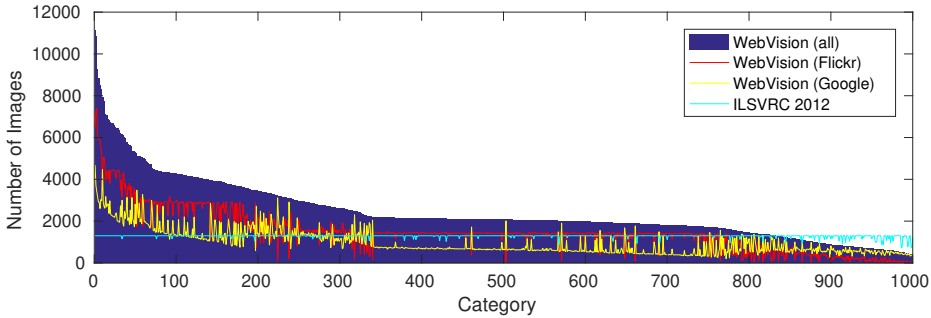

Figure 5: Number of images per category of the WebVison dataset [25].

## B.1 Preprocessing

All images of each dataset are normalized and augmented by random horizontal flipping. For CIFAR-10/100, we use the standard $32 \times 32$ random cropping after zero-padding with 4 pixels on each side. In order to adapt to the input size of CLIP, we upsample images of CIFAR using *torch.nn.functional.interpolate* in PyTorch. For WebVision, the images are initially resized to a uniform size of 256. Subsequently, standard data augmentation techniques are applied, involving the random cropping of patches with dimensions of $224 \times 224$ from each image, followed by the application of horizontal random flipping.

Table 6: Epochs required to reach a target test accuracy on the *100%* WebVision training set (final accuracy is reported in parentheses).

| Dataset | Validation set | Target Acc | 50% data | 100% data |
|---------|---------------|-----------|----------|-----------|
| WebVision-100 | WebVision | 30% | 25 | 12 |
| | | 40% | 40 (60%) | **22** (**68**%) |
| | ILSVRC12 | 30% | 33 | **18** |
| | | 40% | 50 (55%) | **27** (**64**%) |
| WebVision-200 | WebVision | 30% | 22 | **11** |
| | | 40% | 36 (61%) | **18** (**68**%) |
| | ILSVRC12 | 30% | 29 | **16** |
| | | 40% | 46 (56%) | **22** (**64**%) |

Table 7: The results of RHO-LOSS [31] using zero-shot predictor as the validation model on CIFAR-100.

| Method | Target Acc | | Final Acc (%) |
|--------|-----------|--------|---------------|
| | 40.0% | 52.5% | |
| RHO-LOSS w/ zero-shot predictor (CLIP-RN50) | 59 | 92 | 58 |
| RHO-LOSS w/ zero-shot predictor (CLIP-ViT-B/16) | 53 | 86 | 60 |
| Proposed | 32 | 53 | 63 |

**B.2 Hyper-parameters The uning**

We split the original training set into training and validation sets, where the latter remains clean and balanced for hyper-parameters tuning. In fact, as shown in Figure 4, the trade-off coefficient $\alpha$ in the selection objective is the primary factor that impacts the training curve and should be carefully selected. In particular, we select it from $\{0.1, 0.2, 0.3, 0.4\}$ using a small validation set (of size 500 on CIFAR). We reuse the selected $\alpha$ on WebVision-100 without tuning.

**B.3 Network**

For all experiments, ResNet18 models are trained from scratch using PyTorch 2.0.0. Notably, in the case of CIFAR-10/100, we employ a downsampling layer with a small convolution with $3 \times 3$ kernel. Additionally, the average pooling at the end of the network is removed. Following the set-up of BatchNorm in [31], we compute the BatchNorm statistics on large batch $B_t$ for data selection. For model parameters update, we compute the statistics on the selected batch $b_t$.

## C   More Results

Table 6 reports the results of our method on the entire training set of WebVision (as mentioned in Section 4, we use only half of the training set for training in the main experiments for a fair comparison with RHO-LOSS [31]). We see more significant speedups and higher final accuracy due to more training data.

We establish an extra baseline that combines RHO-LOSS with zero-shot predictors. The corresponding results based on our codebase are listed in Table 7. These indicate that the Bayesian treatment introduced by our method plays an important role in accelerating and improving the model convergence.

