# OpenReview forum: "Towards Accelerated Model Training via Bayesian Data Selection"
_NeurIPS.cc/2023/Conference — NeurIPS 2023 poster_

### Official Review · Reviewer_UrxY · 2023-07-06

**Soundness:** 3 good
**Presentation:** 3 good
**Contribution:** 3 good
**Rating:** 5
**Confidence:** 3

**Summary:**

This paper followed up reducible hold-out loss selection (RHO-LOSS) method for data selection and improved it by proposing a reasonable approximation for the non-trivial objective function and eliminating the need for extra hold-out data.

**Strengths:**

1. Authors present both theoretical results (the lower bound for hard-to-estimate objective) and empirical results (algorithm and experiment results).

**Weaknesses:**

1. I am not fully convinced by the paper that their proposed method out-performs existed SOTA, namely RHO-LOSS much, specifically from the results in table 1 and figure 1. I wonder if authors can provide more intuitive comparisons between the proposed method the RHO-LOSS, either empirically show effectiveness and efficiency of the proposed method, or theoretically show the proposed lower bound is more solid than approximation used by RHO-LOSS.
2. The ablation study needs more experiment result (more datasets and more model structure) to get valid conclusion. I also wonder why compare with baseline instead of RHO-LOSS.


**Questions:**

In addition to questions addressed in weakness section, I have following questions for the authors:
1. How is target accuracy selected? Specifically for table 3, why select 30% and 40% as target accuracy?
2. I am curious if each training epoch for RHO-LOSS and your proposed method take same amount of time considering they have different data selection steps?
3. It seems the proposed method performs well on classification datasets with large number of classes (CIFAR 100, ILSVRC12 and WebVision datasets) compared with dataset with smaller number of classes (CIFAR 10). I wonder authors can share some ideas from the perspective of the designed algorithm about this.
4. I wonder if the authors have look into how zero-shot predictor affects the method performance. In other word, is the proposed method robust to poorly-performed predictor? If not, how much will the model be impacted?


**Limitations:**

Yes, authors have addressed their limitation, namely the method performance hugely depends on effectiveness of the zero-shot predictor.

---

> ### Author Rebuttal · Authors · 2023-08-09
>
> We thank you for providing valuable comments. Below, we address each concern in detail, and we sincerely hope that our response proves satisfactory and leads to a higher score.
>
> **Q1: More intuitive comparisons between the proposed method and RHO-LOSS**
>
> **A1:** We first clarify that theoretically, our method builds a more reliable approximation to the selection objective in Eq 4 than RHO-Loss. In particular, RHO-LOSS proposes approximating $\log p(y|x, D^∗, D_{t−1})$ with $\log p(y|x, D^∗)$ and approximating the posterior predictive with the point-estimate models’ predictions, which are both less principled. In comparison, we maintain a Bayesian perspective and develop a lower bound, i.e., Eq 9. Our selection objective is intriguing for several reasons: 1) the first term and $\alpha$ times the third term perform expectation and logarithm operations in reverse order, resulting in a quantity similar to the uncertainty estimates in Bayesian deep learning [39]; 2) the third term helps to prevent the selection of redundant samples; 3) the second term prioritizes points with high semantic alignment with their annotations since the validation data follows the ground truth data generating distribution. These three forces are integrated adaptively to accelerate model training across various stages.
>
> In the aspect of empirical comparison, apart from the results reported in Tab. 1-3, the results in Fig. 1 are insightful. Specifically, Fig. 1a evidences that data selected by our method has a lower label noise. Fig. 1b shows that our method selects slightly fewer redundant data points than RHO-LOSS.
>
> We will add these clarifications to the revision and respectfully ask the reviewer to re-evaluate our paper.
>
> **Q2: Regarding ablation study**
>
> **A2:** We provide clarification to the raised concerns. Firstly, our experiments in sections 4.1-4.4 have already demonstrated the effectiveness of our method across various datasets, including normal, noised, imbalanced, and web-scraped data. Therefore, our ablation studies do not specifically focus on this aspect. Secondly, we have conducted ablation studies on both the architecture of the trained model (Fig. 2) and the architecture of the zero-shot predictor (Fig. 3). These ablation studies have already established the generality and versatility of our method.
>
> Since we have already validated the superiority of our method over RHO-Loss in the main experiments and the purpose of the ablation studies is to gain a better understanding of the behavior of our approach, there is no need to compare with RHO-Loss in the ablation.
>
> To summarize, our empirical evaluations and ablation studies provide substantial evidence to support the effectiveness of our method, so we respectfully disagree with the assessment that our current conclusion lacks proper support.
>
> **Q3: How is target accuracy selected?**
>
> **A3:** We follow RHO-LOSS [31] to select the target accuracy on CIFAR. In the case of WebVision, which is not covered by [31], we select 30% and 40% according to the final accuracies of RHO-Loss ($\approx$ 40%) and those of our approach ($\approx$ 60%).
>
> **Q4: Per-epoch training time**
>
> **A4:** Thanks for the suggestion. Here we provide a direct comparison of per-epoch training time between RHO-LOSS and our method on CIFAR-100:
>
> |Method          | Per-epoch training time |
> | -------- | -------- |
> | Uniform  | 14s       |
> | RHO-LOSS  | 18s       |
> | Proposed | 21s       |
>
> As shown, our approach has only slightly increased per-epoch time over RHO-Loss. It arises from that we use a CLIP-ResNet50 zero-shot predictor to compute the second term in Eq 16, whereas [31] uses a ResNet18 validation model. In fact, this gap can be reduced by a simple implementation trick—pre-computing the CLIP-ResNet50/ResNet18 predictions for each sample in the training set before training (this is done in [31] but not employed in our reproduction).
>
> According to Tab. 1, compared to RHO-Loss, we can reduce the number of epochs required to reach 40% accuracy from 48 to 32, and for 52.5% from 77 to 53. Combining these results, we see a $48 * 18 / 32 / 21 = 1.29$ or $77 * 18 / 53 / 21=1.25$ times practical acceleration.
>
> **Q5: Clearer performance gain on datasets with a larger number of classes**
>
> **A5:** This is because the difficulty of the classification problem is directly correlated with the number of classes involved. This is evident in the final accuracy, where achieving approximately 90% accuracy on CIFAR-10 is relatively straightforward while obtaining 60-70% accuracy on the other datasets poses a greater challenge. So, the conclusion is that our method surpasses the baselines more obviously on more difficult problems. The WebVision dataset setting closely resembles real-world scenarios, and the substantial performance improvement demonstrated by our method over the baselines on it validates our practical value.
>
> **Q6: Is the proposed method robust to poorly-performed predictor?**
>
> **A6:** Yes. As validated in Fig. 3 and stated in L267-269, although the zero-shot accuracy of the CLIP-RN50 variant is significantly lower than that of CLIP-ViT-B/16, the speedup effect of the two zero-shot predictors is similar. Note that CLIP-RN50 is the smallest model in the CLIP family, while CLIP-ViT-B/16 is rather large. This finding suggests the robustness of our method against the selection of the zero-shot predictor.
>
> We further try to replace the zero-shot predictor in our method with the validation model in RHO-Loss. The results on CIFAR-100 are listed below:
>
> |Method          | Epochs to reach  40.0% |Epochs to reach  52.5% |Final acc. |
> | -------- | -------- |-------- |-------- |
> | RHO-Loss|    48  |     77|      61|
> | Proposed - zero-shot predictor + validation model from RHO-Loss  |   30    |  52    |   63   |
> | Proposed |    32   |     53|      63|
>
> The above comparison double confirms the robustness of our method and directly proves the Bayesian treatment in our method is beneficial.

---

> > ### Comment · Reviewer_UrxY · 2023-08-16
> >
> > I thank the authors for their detailed responses to my reviews. After carefully reading their response, I decide to raise my score to 5.

---

> > > ### Author Response · Authors · 2023-08-16
> > > **Thanks**
> > >
> > > Thank you for the raised score. We will carefully revise our paper to include the discussions in the rebuttal. Thank you again!

---

### Official Review · Reviewer_sAWs · 2023-07-08

**Soundness:** 3 good
**Presentation:** 3 good
**Contribution:** 3 good
**Rating:** 7
**Confidence:** 4

**Summary:**

The paper builds on recently proposed work that accelerates training through online batch selection using generalisation loss as selection criterion, by using LaPlace approximation for a stronger Bayesian approximation and using off-the-shelf pre-trained models. The paper presents theory deriving their selection function and then demonstrating its efficacy on a number of datasets in the vision setting.

**Strengths:**

- Novelty on top of RHO-Loss: (1) using pre-trained models as validation model, (2) LaPlace approximation for stronger Bayesian approximation than point-estimate models.
- Well-written paper, that is clean and easy to read.
- Good ablations for how to tune the introduced zero-shot predictor as a validation model.
- Decent performance gains over baselines.

**Weaknesses:**

- The novelty over RHO-Loss is appreciated, but it should be corrected in the narrative that RHO-Loss does not require access to clean holdout data. This is incorrect as far as I can tell, and Mindermann et al. note this in their paper.
- Would be nice to see additional datasets and settings as per the original paper (Language and Clothing1M, however WebVision seems to be a bigger/better dataset)
- In the practical acceleration/algorithm section they note that it's not much slower than [31]. I think a more empirical measurement of the difference would be appreciated, further more how much time does this method add over normal training? Does it really accelerate training in terms of time?
- It's not clear whether the zero-shot predictor or the laplace approximation is providing the gain? An ablation here seems necessary. If it's just the zero-shot predictor, it's seems like there's not much of a difference here besides slapping a pre-trained model on as a validation model?

**Questions:**

As per above.

---

> ### Author Rebuttal · Authors · 2023-08-09
>
> We appreciate the acknowledgment of the novelty of our method in comparison to RHO-Loss, as well as its good performance. Below, we provide detailed responses to the specific comments, hoping that you find them satisfactory and raise your score accordingly.
>
> **Q1: RHO-Loss does not require access to clean holdout data**
>
> **A1:** We apologize for the lack of rigor in our previous statements. We have incorrectly understood that the validation model in RHO-Loss relies on *clean* holdout data (especially in the scenarios with label noise). We sincerely appreciate the reviewer for bringing this to our attention, and we will make the necessary revisions to our paper accordingly.
>
> The main text of [31] presents that RHO-Loss requires training validation models on a separate set of holdout data. We also note that RHO-Loss can work by splitting the training set into two halves and training a validation model on each half to score samples in another half. However, we argue that training additional validation models can be costly and should be performed repeatedly for new tasks.
>
> We assure you that the majority of the arguments presented in our original manuscript remain valid, and will add the above clarifications in the next version.
>
> **Q2: Additional datasets and settings as per the original paper ((Language and Clothing1M)**
>
> **A2:** As mentioned by the reviewer, the used WebVision dataset is more challenging than Clothing1M, so the current empirical study in this regard is convincing. On the other hand, we acknowledge that tasks such as CoLA and SST-2 are relatively straightforward in the field of NLP. Considering recent advancements, such as the GPT and LLaMA series of models for zero-shot recognition, we have not yet conducted experiments on these tasks. We understand that including empirical studies on these tasks is important, so we plan to include them in our final version due to the time constraints during the rebuttal period. Furthermore, we want to emphasize that our current empirical evaluations strongly support the superiority of our method over RHO-Loss.
>
> **Q3: How much time does this method add over normal training? Does it really accelerate training in terms of time?**
>
> **A3:** Thanks for the advice. The wall clock time is indeed an important evaluation metric. Here we provide a direct comparison of per-epoch training time between RHO-LOSS and our method on CIFAR-100:
>
> |Method          | Per-epoch training time |
> | -------- | -------- |
> | Uniform  | 14s       |
> | RHO-LOSS  | 18s       |
> | Proposed | 21s       |
>
>
> As shown, our approach has only slightly increased per-epoch time over RHO-Loss. It arises from that we use a CLIP-ResNet50 zero-shot predictor to compute the second term in Eq 16, whereas [31] uses a ResNet18 validation model. In fact, this gap can be reduced by a simple implementation trick—pre-computing the CLIP-ResNet50/ResNet18 predictions for each sample in the training set before training (this is done in [31] but not employed in our reproduction).
>
> According to Table 1, compared to RHO-Loss, we can reduce the number of epochs required to reach 40% accuracy from 48 to 32, and for 52.5% from 77 to 53. Combining these results, we see a $48 * 18 / 32 / 21 = 1.29$ or $77 * 18 / 53 / 21=1.25$ times practical acceleration.
>
> **Q4: It's not clear whether the zero-shot predictor or the Laplace approximation is providing the gain**
>
> **A4:** Thanks for the constructive comment. We clarify that both the zero-shot predictor and the Bayesian treatment contribute to the success of our method. These factors are integrated into Eq 9 and are traded off by the coefficient $\alpha$. Comparing the curves corresponding to $\alpha=0$ and $\alpha=0.2$ in Figure 4a, we witness that using only the zero-shot predictor is suboptimal and would lead to degraded final accuracy (as stated in L275). On the other hand, too large $\alpha$ (i.e., lightening the second term in Eq 9) can lead to a significant drop in training speed and final performance, which emphasizes the importance of the zero-shot predictor.
>
> We have also ablated on the zero-shot predictor in Figure 3, which proves the robustness of our method against it.
>
> We further try to replace the zero-shot predictor in our method with the validation model in RHO-Loss. The results on CIFAR-100 are listed below:
>
> |Method          | Epochs to reach  40.0% |Epochs to reach  52.5% |Final acc. |
> | -------- | -------- |-------- |-------- |
> | RHO-Loss|    48  |     77|      61|
> | Proposed - zero-shot predictor + validation model from RHO-Loss  |   30    |  52    |   63   |
> | Proposed |    32   |     53|      63|
>
> The above comparisons with RHO-Loss confirm the necessity of introducing the Bayesian treatment.

---

> > ### Comment · Reviewer_sAWs · 2023-08-18
> > **Response**
> >
> > Q1-Q3. Noted.
> >
> > Q4. So perhaps I'm slightly confused here, but it seems like all of the gain is from subbing in a stronger model (zero-shot predictor), nothing from what is actually proposed in this paper (in fact it loses performance in two cases shown). From RHO-LOSS paper this seems to be an obvious perhaps even implied step?
> > If my understanding is incorrect, then the experiment I asked for remains: RHO-LOSS with CLIP/zero-shot predictor vs RHO-LOSS and proposed model.

---

> > > ### Author Response · Authors · 2023-08-18
> > > **Thanks for your reply**
> > >
> > > We apologize for any confusion caused by our previous reply. We would like to provide further clarification regarding the method **Proposed - zero-shot predictor + validation model from RHO-Loss**. This method specifically refers to the variant of our method that utilizes the validation model of RHO-Loss to **take the place of the zero-shot predictor**. It is important to note that the key distinction between this method and RHO-Loss lies solely in the selection principle. The significant disparities in training speed and final performance serve as evidence that our selection principle is indeed effective.
> > >
> > > We acknowledge that including a baseline that combines RHO-Loss with a zero-shot predictor could be beneficial in double-checking this point. The corresponding results of RHO-Loss based on our codebase are listed below:
> > >
> > > |Method          | Epochs to reach  40.0% ACC $\downarrow$ |Epochs to reach  52.5% ACC $\downarrow$ |Final ACC. (%) $\uparrow$ |
> > > | -------- | -------- |-------- |-------- |
> > > | RHO-Loss w/ zero-shot predictor (CLIP-RN50)|  59  |   92 |  58 |
> > > | RHO-Loss w/ zero-shot predictor (CLIP-ViT-B/16)|  53  |  86  |  60|
> > > | Proposed |    32   |     53|      63|
> > >
> > > Given these, we would like to ask the reviewer to re-evaluate our contributions. We welcome any further comments.

---

> > > > ### Comment · Reviewer_sAWs · 2023-08-18
> > > > **Response**
> > > >
> > > > Fantastic, this resolves my concerns. I'll be updating my score to 7, great work!

---

> > > > > ### Author Response · Authors · 2023-08-18
> > > > > **Thanks**
> > > > >
> > > > > Thank you very much! We will carefully revise our paper to incorporate the discussions and results in the rebuttal. Thank you again!

---

> > > > > ### Comment · Reviewer_sAWs · 2023-08-18
> > > > > **One other concern**
> > > > >
> > > > > Just a brief lookover again. Are these the results of a single seed? It seems like that is the case, as only in figure 1 is it reported that multiple runs are used. Multiple runs should be used in evaluating the results as these algorithms are not the most robust from my experience.

---

> > > > > > ### Author Response · Authors · 2023-08-19
> > > > > > **Regarding random runs**
> > > > > >
> > > > > > Sorry for the lack of clarity. The results in the tables of our paper correspond to the average over 3 random runs. The results in the rebuttal are from a single seed and we did not engage in cherry-picking. We have empirically observed that our performance remains relatively stable across different runs. We will be careful about this point when revising our paper. Thanks for your kind advice!

---

### Official Review · Reviewer_GG6S · 2023-07-27

**Soundness:** 3 good
**Presentation:** 3 good
**Contribution:** 3 good
**Rating:** 6
**Confidence:** 4

**Summary:**

This paper studies data selection methods because training examples may be of different importance/quality. By selecting a subset of high-quality/high-usefulness examples, the model’s performance can be improved when training on this subset. In Particular, the paper proposed a method by leveraging a light-weight Bayesian treatment and incorporating off-the-shelf zero-shot predictors.

**Strengths:**

1. The paper is clearly motivated.
2. The proposed method is well explained.
3. The experiments cover different baselines, models and datasets.


**Weaknesses:**

The problem of data selection has been studied extensively in the ML community, which verifies the importance of this problem. Some recent works ([1,2] from reference list below) proposed data selection methods that do not only focus on hard/easy examples, but also consider data distribution and its impact on the overall loss (e.g., see Theorem 1 from [2]). It looks like these methods may already have partially addressed the problem studied in this paper. It would be better if the authors could include some discussion about these methods and possibly compare against them in the experiments.

Reference:
[1] Xia, Xiaobo, et al. "Moderate coreset: A universal method of data selection for real-world data-efficient deep learning." ICLR 2023.
[2] Zheng, Haizhong, et al. "Coverage-centric Coreset Selection for High Pruning Rates." ICLR 2023.

**Questions:**

Please see my comments in Weaknesses

---

> ### Author Rebuttal · Authors · 2023-08-09
>
> We thank you for your supportive reviews and for finding our work clearly motivated and well-explained. We address the detailed concerns below.
>
> **Q1: Some discussion and comparison with recent works**
>
> **A1:** Thanks for the constructive suggestion. The problem of data selection is indeed important and well-studied in the ML community, but we clarify that our method makes novel, valuable contributions. In particular, the mentioned papers operate in a different setting from our work: they focus on coreset selection while we perform the online batch selection. More specifically, reference [1] contributes a data scoring mechanism that is robust to the change of scenarios for coreset selection, and reference [2] makes an in-depth understanding of the catastrophic accuracy drop issue of one-shot coreset selection and contributes a novel solution to it. Compared to them, our method selects valuable samples based on their marginal influence on the model’s generalization loss during training, so samples with different properties can be selected at various training stages. As a result, we can boost not only the final accuracy but also the training speed. This can be a major difference. Moreover, our selection is dependent on the model in training, while the mentioned two papers are not, so these approaches are expected to handle various scenarios in practice. Anyway, we will add more thorough discussions of these works in the revision and attempt to include direct empirical comparisons.

---

### Official Review · Reviewer_agRJ · 2023-07-29

**Soundness:** 3 good
**Presentation:** 3 good
**Contribution:** 3 good
**Rating:** 7
**Confidence:** 4

**Summary:**

This works is situated in the field of **robust generalization**; in particular it studies the problem of how models that were trained on noisy or imbalanced data perform on clean data. They achieve this via **online batch selection**, and in particular they contribute to the development of a **Bayesian framework** for searching for the best datapoints to use in each mini-batch among some canditate points sampled from the training set. In more details, they extend a SotA method, called **RHO-LOSS** [1]. The authors pinpoint some approximation choices that [1] makes in order to make the principled Bayesian approach to data selection practical. As a result, they design an algorithm which, in contrast to [1], they claim 1) to make **less crude approximations about predictive posteriors**, by using variational Laplacian approximations, MC sampling (and other) instead of pure point estimates, and 2) **not to rely on the existence of oracle clean data** during the training of their model. Their experiments demonstrate **improvements in robust generalization and training acceleration for image classification tasks** over other methods in the literature (incl. [1]).

[1] Mindermann, Sören, et al. "Prioritized training on points that are learnable, worth learning, and not yet learnt." International Conference on Machine Learning. PMLR, 2022.

**Strengths:**

The authors contribute to an important direction for robust training methods, in particular developping a bit further the Bayesian data selection framework.
1.  The paper is well-written and motivated. The derivations are easy to follow and the deferrals to the appendix are used appropriately. Presentation can further be improved if there was somewhere a paragraph (perhaps in the appendix) with the list of approximation steps taken (for future reference), e.g. MC sample of a variational posterior, based on Laplace approximation of the last layer of a neural network (for which the backbone is taken as a point estimate), and the Laplace approximation further does not use the true Gaussian, but a Gauss-Newton matrix + potentially a KFAC approximation of it.
2.   Authors identify correctly potential improvement points of [1], and proceed in mitigating them. In this way, they introduce a novel approximation scheme to the Bayesian framework of data selection, which is effective and computationally-lightweight.
3.   Experiments seem to advocate in favor of using their method over [1] or other methods, they are extensive and on various dataset scales in image classification tasks.

**Weaknesses:**

1.   Proposed method overly claims that it does not use oracle clean data, however it might still depend on clean data via the unsupervised pre-trained zero-shot classification proxy that they use. In particular, CLIP-R50 might have been trained in a superset of datasets (like ImageNet) which contains CIFAR10/100. In that case, information from oracle clean data has been stored in the pretrained model. While I believe that this is a weaker assumption, there needs to be an explicit statement of this assumption and a quantitative assessment of this potential pitfall. To which extent would a pretrained model on the same noisy/imbalanced dataset as the benchmark task be useful? Nonetheless, the authors acknowledge this potential limitation of an underperforming pretrained model at the last section. The following (lightweight) experiments can be further performed to augment their arguments or awareness of the limitation:

 -  A zero-shot CLIP baseline is provided, I think they should also provide with a finetuned version using linear probing and training on imbalanced/noisy CIFAR10/100 with uniform sampling. This way it will be more clear that the improvement are more due to their method and not because of an overpowered pretrained model.
 -  Consider CIFAR10/100 experiments using an unsupervised pretrained model, with linear probing or kNN (so that it is zero-shot if you want that) for the oracle model, via for instance some simple SSL method like MoCov2 [2] on the same noisy/imbalanced training set.

I will increase the assessed score if such experiments are performed and reported.

2.   Ablation study reveals sensitivity to some hyperparameters. As a result, model selection and hyperparameter tuning are important for the success of the method. How was model selection performed? Was the validation split iid to the training set or was it a clean/balanced dataset? This needs to be reported clearly, but not addressed in this paper.

3.   It would be nice to have a more extensive ablation study on the effects of the approximation choices in terms of final test accuracy, spanning from crude approximation schemes to the one finally used by the authors.

[2] Chen, Xinlei, et al. "Improved baselines with momentum contrastive learning." arXiv preprint arXiv:2003.04297 (2020).

**Questions:**

### Questions

* (related to weakness 2) How was model selection performed?
* How were target accuracies chosen for training acceleration results?
* Lines 151-152: What do we miss by replacing the Hessian matrix in the Laplacian approximation by the Gauss-Newton?
* Lines 131-132 about “the recent trend of exploring the potential of pretrained models” (in robust generalization) needs some citations.
* What does Table 4 in the Appendix refer to? Please complete the appendix to explain.

### Typos

* y-axis on Fig.2,3 and 4 are not in % of test accuracy as indicated.

**Limitations:**

The authors derive a novel approximation scheme to Bayesian data selection for training robust models (Strength 2), with favorable results over past literature in image classifiacation tasks (Strength 3) and the writing is overall great (Strength 1). There is a concern about the applicability of the method in cases where a pretrained model does not exist for the task at hand, in which case we need to find another proxy or probably pretrain it (Weakness 1), and how much actually the method makes no use of priviledged clean/balanced datasets for training (Weakness 1) and validation (Weakness 2). The limitation about the pretrained model is mentioned briefly, but it would be appreciated if it is expanded and experimented on with the ablation studies suggested in the Weakness section above.

---

> ### Author Rebuttal · Authors · 2023-08-09
>
> We appreciate your positive review and recognition of the presentation, novelty, and effectiveness of this work. Below, we provide answers to the specific questions raised.
>
> **Q1: To what extent does the effectiveness of this work stem from the effectiveness of the CLIP-based zero-shot predictor that was used?**
>
> **A1:** Thanks for the comments and suggestions. We make the following clarification.
>
> Firstly, we clarify that the zero-shot predictor used in our work contains valuable information regarding CIFAR while *not* being trained on a superset of CIFAR like ImageNet. Specifically, CLIP has been trained on a web-scale collection of image-text pairs using an *aligning loss*. Secondly, we emphasize that large-scale pre-trained models like CLIP, BLIP, SWAG, and others have become essential infrastructure for modern AI. These models are openly available, freely accessible, and generally applicable. In this regard, our method is more favorable than RHO-Loss [31].
>
> As mentioned in our paper and noticed by the reviewer, our method does not hinge on a performant zero-shot predictor. For example, in Tab. 1, a CLIP zero-shot predictor with 75.6% accuracy on CIFAR-10 leads to a final accuracy of 91% for our method. As shown in Fig. 3, the speedup effect of CLIP-RN50 is similar to that of CLIP-ViT-B/16 on CIFAR-100, while the former is significantly weaker.
>
> We comment that we use the zero-shot predictors directly without tuning in all the experiments reported in our paper. We conduct a set of experiments following the reviewer’s suggestion—linear probing using CLIP models on the clean/noisy CIFAR-10/100 with uniform sampling. The table below displays the comparison of the final accuracy (the comparison on convergence speed is not sensible because this baseline uses pre-trained weights while the methods listed in Table 1&2 have not):
>
> |   Method/dataset                                   | CIFAR-10 | CIFAR-10* | CIFAR-100 | CIFAR-100* |
> | ------------------------------------ | -------- | --------- | --------- | ---------- |
> | Linear probing with uniform sampling | 84.5     | 84.1      | 58.5      | 57.8       |
> | Proposed                             | 91.4     | 91.3      | 63.3      | 61.4       |
>
>  We can see that the proposed method outperforms linear probing using CLIP on both clean (CIFAR-10/100) and data with 10% symmetric label noise (CIFAR-10*/100*).
>
> We also add a new experiment where we replace the zero-shot predictor in our method with the validation model used in RHO-Loss. The results on CIFAR-100 are listed below:
>
> |Method          | Epochs to reach  40.0% |Epochs to reach  52.5% |Final acc. |
> | -------- | -------- |-------- |-------- |
> | RHO-Loss|    48  |     77|      61|
> | Proposed - zero-shot predictor + validation model from RHO-Loss  |   30    |  52    |   63   |
> | Proposed |    32   |     53|      63|
>
> The above comparison confirms the necessity of introducing the Bayesian treatment and highlights the superiority of our method over RHO-Loss.
>
> **Q2: How was model selection performed?**
>
> **A2:** We split the original training set into training and validation sets, where the latter remains clean and balanced for model selection.
> In fact, as shown in Figure 4, the trade-off coefficient $\alpha$ in the selection objective is the primary factor that impacts the training curve and should be carefully selected. In particular, we select it from $\{0.1, 0.2, 0.3, 0.4\}$ (see L207) using a small validation set (of size 500 on CIFAR). We reuse the selected $\alpha$ on WebVision-100 without tuning.  We’ll make these points clearer.
>
> **Q3: A more extensive ablation study on the effects of the approximation choices in terms of final test accuracy**
>
> **A3:** Thanks. We will add a more in-depth empirical analysis of our method in the revision. The core of our method lies in the computation of $\log p(y|x, D^∗, D_{t−1}) - \log p(y|x, D_{t−1})$. In particular, we approximate $\log p(y|x, D^∗, D_{t−1})$ with its lower bound for tractability, and approximate the predictive built on validation data with zero-shot predictors. These two choices are our major technical contributions and differences from RHO-LOSS [31]. The former cannot be trivially ablated and the latter has been well studied by us. For Bayesian treatment, we deploy online Laplace approximation for the posterior update, GGN approximation to avoid ill-posed Hessian, and last-layer KFAC approximation for tractability. These choices are essential. For example, introducing variational inference or MCMC for posterior inference will cause substantially higher costs and implementation challenges.
>
> In fact, RHO-LOSS [31] is a counterpart of our approach that uses less principled and less reliable approximations. This is verified by our existing experiments.
>
> **Q4: How were target accuracies chosen for training acceleration results?**
>
> **A4:** Those in regular and noisy-label experiments follow RHO-LOSS [31]. Those in imbalanced experiments are selected according to the final accuracies achieved by our method and the baselines.
>
> **Q5: What do we miss by replacing the Hessian with Gauss-Newton?**
>
> **A5:** While the theory advocates using Hessian for Laplace approximation, practical experience suggests substituting it with the Gauss-Newton matrix [8, 37]. Dosing so, the information in the probability curvature mostly remains. Besides, the Gauss-Newton matrix is positive semi-definite and can be more easily manipulated in practice.
>
> **Q6: Missed citation in L131**
>
> **A6:** Thanks for the suggestion. We’ll fix this issue in the next version.
>
> **Q7: What does Table 4 in the Appendix refer to?**
>
> **A7:** It reports the results of our method on the *entire* training set of WebVision (see L195: we use only half of the training set for training in the main experiments for a fair comparison with RHO-LOSS [31]). We see more significant speedups and higher final accuracy due to more training data.

---

> > ### Comment · Reviewer_agRJ · 2023-08-11
> > **Thank you for your responses**
> >
> > **Q1**: Regarding the point of “superset” dataset, the web-scale collection of image-text pairs almost certainly is exposed to images of CIFAR10 categories (airplanes, cars, birds, cats, deer, dogs, frogs, horses, ships, and trucks). The authors can try to disprove this. The more CLIP, BLIP and SWAG are trained in larger-scale in-the-wild datasets, the more difficult it is to claim zero-shot generalization to ood data. This is not up to the authors to disprove/approve, they just have to provide a convincing enough ablation, that in light of this their method still provides with further benefits.
> >
> > Thank you for the linear probing experiments, I would have been more convinced if you provided a *finetuned version* of the R50-CLIP network instead.
> > The second experiment, however, provides with enough ablation against RHO-Loss.
> >
> > **Q2**: Please be clear about the validation protocol, it is important to clarify (if not address with the same resources as training).
> >
> > **Q7**: Please clarify in the appendix as well Table 4, since there is no text accompanying it.
> >
> > Thank you for answering the rest of my questions, I raise my score to 7.

---

> > > ### Author Response · Authors · 2023-08-13
> > > **Thank you for your feedback**
> > >
> > > We appreciate your detailed comments and suggestions.
> > >
> > > Regarding the zero-shot predictor, we totally understand what you're trying to convey. As suggested, we provide a further baseline—the finetuned version of the R50-CLIP, with the results detailed below:
> > > |                                      | CIFAR-10 | CIFAR-10* | CIFAR-100 | CIFAR-100* |
> > > | ------------------------------------ | -------- | --------- | --------- | ---------- |
> > > | Fine-tuning R50-CLIP with uniform sampling | 87.6    |  85.3    | 59.1     |   57.6     |
> > > | Proposed                             | 91.4     | 91.3      | 63.3      | 61.4       |
> > >
> > > We will incorporate your suggestions on validation protocol and Table 4 in the final revision and continually polish our paper. Thank you again!

---

### Decision · Program_Chairs · 2023-09-21

**Decision:**

Accept (poster)

**Comment:**

The work provides a Bayesian framework for online batch selection optimizing for generalization loss. The authors improve upon previous works by using a more precise approximation and using less assumptions. The reviews agree that the problem is interesting, and that the provided technique is convincing both in terms of having theoretical backing and in terms of having thorough experiments showcasing its advantage over previous art. Some issues around clarity were raised by the reviews but they were resolved in the discussion phase. I think the paper will be a great addition to Neurips and urge the authors to carefully go through the reviewer comments and add the necessary changes towards the paper’s final version.